# Cellular and Genomic Instability Induced by the Herbicide Glufosinate-Ammonium: An In Vitro and In Vivo Approach

**DOI:** 10.3390/cells13110909

**Published:** 2024-05-24

**Authors:** Alfredo Santovito, Mattia Lambertini, Dáša Schleicherová, Enrico Mirone, Alessandro Nota

**Affiliations:** 1Department of Life Sciences and Systems Biology, University of Turin, Via Accademia Albertina 13, 10123 Torino, Italy; alfredo.santovito@unito.it (A.S.); dasa.schleicherova@unito.it (D.S.); 2Department of Chemistry, University of Turin, Via P. Giuria 7, 10125 Torino, Italy; mattia.lambertini@edu.unito.it; 3Department of Biosciences and Territory, University of Molise, Via Francesco De Sanctis 1, 86100 Campobasso, Italy; e.mirone@studenti.unimol.it; 4Department of Biology and Biotechnology, University of Pavia, Via Ferrata 9, 27100 Pavia, Italy

**Keywords:** micronuclei, nuclear buds, genotoxicology, organophosphate, ecotoxicology

## Abstract

Glufosinate-ammonium (GLA), an organophosphate herbicide, is released at high concentrations in the environment, leading to concerns over its potential genotoxic effects. However, few articles are available in the literature reporting the possible cellular and nuclear effects of this compound. We assessed, by in vitro and in vivo micronucleus assays, the genotoxicity of GLA on cultured human lymphocytes and *Lymnaea stagnalis* hemocytes at six concentrations: 0.010 (the established acceptable daily intake value), 0.020, 0.050, 0.100, 0.200, and 0.500 µg/mL. In human lymphocytes, our results reveal a significant and concentration-dependent increase in micronuclei frequency at concentrations from 0.100 to 0.500 μg/mL, while in *L. stagnalis* hemocytes, significant differences were found at 0.200 and 0.500 μg/mL. A significant reduction in the proliferation index was observed at all tested concentrations, with the only exception of 0.010 μg/mL, indicating that the exposure to GLA could lead to increased cytotoxic effects. In *L. stagnalis,* a significant reduction in laid eggs and body growth was also observed at all concentrations. In conclusion, we provided evidence of the genomic and cellular damage induced by GLA on both cultured human lymphocytes and a model organism’s hemocytes; in addition, we also demonstrated its effects on cell proliferation and reproductive health in *L. stagnalis*.

## 1. Introduction

Intensive agriculture practices require the massive use of pesticides to control weeds that can affect the crop productivity and the quality of both human life and the environment. Among pesticides, herbicides are the most applied and, consequently, are found at high concentrations in the environment, particularly in soil and aquatic ecosystems [1]. However, due to their non-specific action, these chemicals can harm non-target organisms like pollinators and aquatic invertebrates with a consequent reduction in biodiversity and non-negligible economic damage due to honey bee colony losses in agricultural ecosystems [2,3].

Glufosinate-ammonium (GLA) is an organophosphate wide-spectrum herbicide that, in plants, exerts its action by irreversible inhibition of the glutamine synthetase (GS) activity, resulting in photosynthesis disruption (for a review, see Takano and Dayan, 2020) [4]. Briefly, the increase in ammonia concentration and glyoxylate derivatives, due to GS blockade, leads to an excess of energy within the cell, which is consumed by electron transfer to molecular oxygen. Reactive oxygen species deriving from this process are the main drivers of GLA phytotoxicity, being able to induce lipid peroxidation, membrane integrity loss [5], and chromosomal instability [6,7].

The use of GLA is significantly increasing worldwide. It is principally used to control weeds in genetically modified crops tolerant to glyphosate [8,9], but its use might have an adverse impact also on non-target plants, such as trees, shrubs, and ornamental crops [10], as well as on the entire biota. Indeed, due to its high solubility in water, GLA has been found in all environmental matrices, including the soil and surface waters [11,12].

The maximum residue limit of GLA in European drinking waters was set at a concentration of 0.100 μg/L [13]. However, in northern Italy, the annual average concentration of GLA in the rivers Musoncello (0.720 μg/L) and Teva (0.420 μg/L) exceeded this upper tolerable limit for Europe [11], and, in the Italian Veneto region, the average concentration of GLA in drinking water reached this maximum allowable limit as well [11,14]. Outside Europe, the situation is similar: GLA was found at concentrations ranging from 2.380 μg/L to 13.150 μg/L in waters and groundwaters of ten provinces of China [15], at concentrations ranging from 0.020 μg/L to 15.500 μg/L in three watersheds in the Midwest of the United States [16]. 

This situation represents a notable risk to aquatic life. Indeed, significant toxic effects on non-target organisms have been observed. In particular, male reproductive and developmental toxicities have been reported in mammals exposed to GLA [17]; morphological abnormalities during early development in amphibians [18]; and oxidative stress, hormonal disturbance, and DNA damage in tadpoles. Moreover, in zebrafish, GLA was found to cause spinal deformities, yolk sac edemas, and embryo mortality [19], whereas, in reptiles, it induces hepatotoxicity and reproductive toxicity [20].

In humans, epidemiological studies demonstrated that the exposure to this pesticide is associated with child neurodevelopmental defects and cognitive disorders [21,22], as well as with respiratory and immunological diseases [23,24]. 

From a cellular perspective, glufosinate can often affect cell integrity and vitality. GLA exposure was found to induce a loss of cell–cell interaction in the ependymal epithelium, undermining the tissue’s integrity [25]. Furthermore, it has been found to adversely affect the subcellular structure of the aquatic unicellular alga *Chlorella vulgaris*, damaging chloroplast structures and reducing photosynthetic activity [26].

In tadpole erythrocytes, GLA already demonstrated its cytotoxic effects [9]. In *Caenorhabditis elegans,* exposure to GLA at environmental concentrations resulted in the inhibition of GS, with a consequent accumulation of glutamine and induction of programmed cell death [27]. Similarly, GLA was found to increase the incidence of apoptosis and to affect the normal in vitro growth of preimplantation embryos, but these effects only occur at high concentrations (15 μg/mL and 150 μg/mL, respectively) [28].

However, data about the genotoxic properties of GLA are scanty and conflicting [29,30]. In the present paper, we evaluated, in vitro on cultured human lymphocytes and in vivo on hemocytes of *Lymnaea stagnalis*, the effects of the exposure to low GLA concentrations. We tested GLA at a concentration of 0.010 μg/mL, corresponding to the established acceptable daily intake (ADI) value for humans [13], and its multiples of 0.020, 0.050, 0.100, 0.200, and 0.500 μg/mL.

The genomic instability was evaluated by the micronucleus assay performed on human lymphocytes and on hemocytes of *L. stagnalis*. The cytokinesis-block micronucleus (CBMN) test was initially developed on human peripheral blood lymphocytes. However, in the ecotoxicological field, this test has also been performed on freshwater invertebrates, such as *L. stagnalis* [31]. The latter is an abundant and widespread invertebrate species that colonizes the limnic temperate systems of the Northern Hemisphere and North Africa. The hemolymph system represents an important membrane for pathogen protection and elimination; in addition, hemocytes play various other functions, including phagocytosis and cell-to-cell communication and recognition [32]. The number of hemocytes is correlated with immunity and is an early indicator of infection in aquaculture conditions [33]. Among the target cells used in the literature to evaluate the effects of exposure to genotoxic agents, lymphocytes represent the ideal biological dosimeters for a variety of reasons: (a) they are cells that circulate throughout the body and, therefore, can be considered circulating dosimeters able to detect the genotoxic insult received from the various districts of the organism; (b) they can be obtained by a common venous sampling and are easy cultured in vitro; (c) finally, the lymphocyte memorizes the damage, and, therefore, if the chromosomal damage persists unrepaired in cells in the G0 phase, the same can be detected when the cell undergoes mitotic division in vitro. Therefore, an increased frequency of MN may be the result of recent exposures, or, in some cases, of previous exposures, and therefore the manifestation of a cumulative effect [34].

*L. stagnalis* is often used as a model organism in ecotoxicological and biological studies, as it has different advantages: (a) It is sexually mature within only 3 months after egg laying. (b) It is a simultaneous hermaphrodite, which means that it can both cross- and self-mate. (c) During mating, partners regularly alternate sexual roles by the reciprocal exchange of sperm. These characteristics offer the possibility to study the influence of xenobiotic compounds on life traits related to the reproductive aspects, such as the number of laid eggs. (d) *L. stagnalis* can be easily maintained in the laboratory; it tolerates a wide range of temperatures (from 0 to 26–28 °C) and pHs (from pH 6 to pH 8.5). (e) This organism does not require a particular diet, because it is omnivorous and feeds mainly on algae and plants. (f) Finally, *L. stagnalis* lives mainly in ponds, where the concentration of xenobiotics is higher than in other aquatic ecosystems. This means that they have probably evolved efficient cellular detoxification mechanisms and, therefore, the genotoxic effect of a certain environmental xenobiotic observed on this organism could have even worse consequences in other aquatic and terrestrial organisms living in contaminated ecosystems [35].

The micronucleus assay represents a fast and inexpensive test able to assess the aneugenic and/or clastogenic properties of a single xenobiotic or a mixture of different compounds. Indeed, micronuclei (MNi) represent whole chromosomes or fragments of them that failed to migrate to the anaphase during mitosis, resulting in visible extranuclear bodies in interphase nuclei. This assay can also measure another nuclear anomaly called nuclear buds (NBUDs), which are produced by eliminating amplified DNA, DNA repair complexes, and excess chromosomes in aneuploid cells [36]. Finally, the CBPI represents a cytokinesis-block proliferation index and is calculated with the following formula, [1 × N1 + [2 × N2 + 3 × (N3 + N4)]/N], where N1–N4 represent the number of cells with 1–4 nuclei, respectively, and N is the total number of cells evaluated [37]. This index gives an idea of the influence of a specific xenobiotic on the cell’s ability to divide: low values of this index correspond to a lower ability of the cell to divide, and this may represent the cellular manifestation of possible damage to the cytokinetic mechanism.

Various pesticides have been reported to lead to a wide range of cellular toxicity mechanisms, including cell cycle arrest, which represents one of the most important toxic consequences of xenobiotics [38,39]. However, little is known about the effects of GLA on cell division rates. For these reasons, we decided to evaluate in our analyses also the proliferation index of lymphocytes treated with different concentrations of GLA. 

## 2. Materials and Methods

### 2.1. Chemicals and Reagents

The IUPAC name of GLA (C_5_H_12_NO_4_P) is ammonium (3-amino-3-carboxypropyl) methyl phosphinate (CAS number: 77182–82-2). GLA, colchicine, Cytochalasin-B, and Mitomycin-C (MMC), KCl, and Sörensen buffer were purchased from Merck S.p.A., Milan, Italy. Foetal calf serum, RPMI 1640 cell culture medium, phytohemagglutinin, and antibiotics were obtained from Thermo Fisher Scientific, (Rodano, Milano, Italy). Methanol, Giemsa, acetic acid, and microscope slides were obtained from Carlo Erba Reagenti (Cornaredo, Milan, Italy). Vacutainer blood collection tubes were obtained from Terumo Europe (Rome, Italy). 

### 2.2. Lymphocyte Cultures and Cytokinesis-Block Micronucleus Assays

Peripheral venous blood was collected from 20 healthy Italian adults (mean age ± SD, 24.000 ± 2.248), with no current tobacco or alcohol consumption, not under any drug therapy, and without any recent history of exposure to mutagens. Signed informed consent was obtained from all subjects. This study was approved by the Ethics Committee, University of Turin (protocol no. 0574348, dated 18 October 2023), and performed in accordance with the ethical standards stated in the 2013 Declaration of Helsinki. 

Blood sample collection, lymphocyte cultures, fixation, staining procedures, a micronuclei assay, and microscope analysis were performed as described in Santovito et al. (2018) [40]. Lymphocyte cultures were treated with six nominal concentrations (0.010, 0.020, 0.050, 0.100, 0.200, 0.500 μg/mL) of GLA. The positive control was treatment with Mitomycin-C (Mit.C) at 0.100 μg/mL. 

The frequencies of micronuclei (MNi) and nuclear buds (NBUDs) were measured in 1000 binucleated lymphocytes with well-preserved cytoplasm per subject, per concentration and per tested compound, for a total of 120,000 scored cells. Similarly, a total of 1000 lymphocytes per donor per concentration were scored to determine the cytokinesis-block proliferation index.

### 2.3. Lymnaea stagnalis Physiology and Rearing

*L. stagnalis* individuals are simultaneous hermaphrodites that alternate sexual roles during mating, with the reciprocal exchange of sperm, i.e., sperm trading [41]. They become sexually mature within 3 months after egg laying, have a lifespan of about 2 years, and can be easily maintained in the laboratory due to their broad temperature tolerance (0 to 26–28 °C) and pH (pH 6 to pH 8.5).

*L. stagnalis* individuals involved in this study were randomly selected from our parasite-free laboratory culture, where rearing conditions were uniform in terms of tap water, temperature (range between 18 and 22 °C), and feeding (salad as the primary source of food). 

The experiment was conducted using 10 L containers filled with 8 L of tap water. To reduce evaporation, the containers were closed with a lid, but air could pass via lateral slits on the top of the containers. The chemistry of the used tap water was pH: 7.4; dry residue at 180 °C: 307 mg/L; calcium: 69 mg/L; magnesium: 13 mg/L; ammonium: <0.05 mg/L; chlorides: 14 mg/L; sulfates: 32 mg/L; potassium: 2 mg/L; sodium: <10 mg/L; and arsenic: <1 µg/L. Glufosinate and other pesticides were not found [42]. A single replicate of 20 individuals per concentration was used, for a total of seven groups with 20 individuals randomly chosen among young sexual mature adults (with a shell length ≥20 mm). We tested the effects of GLA at the genomic level in terms of the variation in the frequency of MNi, and on some physiological parameters, such as body growth and egg production. In order to obtain the final concentrations of 0.500, 0.200, 0.100, 0.050, 0.020, and 0.010 mg/L, we dissolved 4.0, 1.6, 0.8, 0.4, and 0.2 g of GLA in 8 L of water, respectively. 

Individuals were reared at room temperature (range, 18–22 °C), under the same light–dark regime, and fed ad libitum the same amount of food (100 g of vegetables/week for each set of 20 individuals, mainly salad). In order to avoid using salad treated with pesticides, we fed *L. stagnalis* with organic salad. During the 4-week study period, water, food, and GLA concentrations were renewed twice a week for each group. The number of eggs laid was recorded once a week and changes in shell growth (length in mm) were measured at the beginning and at the end of the experiment. 

### 2.4. Micronuclei Assay on Hemocytes from Lymnaea stagnalis

Twenty individuals per group were exposed for 4 weeks to increasing concentrations (0.010, 0.020, 0.050, 0.100, 0.200, and 0.500 μg/mL) of GLA. The negative control was represented by pure water without GLA. At 4 weeks of xenobiotic exposure, hemolymph was collected by stimulating its release by prodding the animal’s foot with a micropipette. Five hundred microliters of hemolymph per subject was collected and distributed onto clear microscope slides. The cells were then fixed by adding several drops of methanol/acetic acid solution in a 3:1 ratio. The slides were dried, and the cells were stained for 10 min by a conventional staining method using 5% Giemsa (pH 6.8) prepared in Sörensen buffer. Then, the slides were washed with distilled water and dried; the cells were observed under a Leica Dialux 20 light microscope (magnification 1000×). We analyzed 1000 hemocytes with intact nuclear and cellular membranes per subject per concentration, and the number of MNi was scored. MNi were identified according to the following criteria: diameter < 1/3 of the main nucleus; coloring and refractive index similar to those of the main nucleus; and absence of direct contact between the MN and the main nucleus.

### 2.5. Statistical Analyses

Data distribution and data normality were tested graphically and using the Shapiro–Wilk test. A one-way ANOVA was used to test for growth differences between the groups, with a Tukey post hoc test. For MNi assays, as the data were not normally distributed (even after data transformation), the non-parametric Kruskal–Wallis test was used to test for significant differences in terms of nuclear aberrations in both lymphocytes and hemocytes, with a post hoc Dunn test (Bonferroni correction). Due to the non-homogeneity of data distribution for CBPI values, a GLM with negative binomial distribution was applied to test for significant differences. Differences in the number of eggs laid between the groups were tested with the G-test. A principal component analysis (PCA) was conducted with the scaled variables “Micronuclei”, “Nuclear buds”, “Binucleated cells”, “Growth”, and “Number of eggs laid”. Statistical significance was indicated as * *p* < 0.05, ** *p* < 0.01, and *** *p* < 0.001. All statistical analyses were performed using R 4.3.2 (R core team, Vienna, Austria) with the Rstudio interface (RStudio Team, Boston, MA, USA). Graphs were elaborated on R and GraphPad Prism 8 (GraphPad Software, https://www.graphpad.com, accessed on 16 March 2024).

## 3. Results

### 3.1. Lymphocytes

In Figure 1 and Appendix A, differences in the levels of micronuclei and nuclear buds in lymphocytes exposed to different concentrations of GLA are highlighted. The analytical data are reported in Appendix A. GLA was genotoxic at concentrations of 0.500, 0.200, and 0.100 μg/mL, whereas no significant differences in terms of buds were found between different GLA concentrations and the negative control. 

In Figure 2 and Appendix A, differences in the CBPI values among lymphocytes treated with different GLA concentrations are shown. With respect to the negative control, the CBPI value was significantly reduced at all concentrations with the exception of 0.010 μg/mL, indicating a negative effect of GLA on the proliferation index at concentrations higher than the ADI value. 

### 3.2. Hemocytes

In Figure 3 and Appendix A the frequencies of MNi and NBUDs in hemocytes exposed to different GLA concentrations are shown. GLA significantly increased the frequencies of MNi and NBUDs at concentrations of 0.500 and 0.200 µg/mL (Figure 3).

In Figure 4 and Appendix A, the variation in shell growth between controls and individuals exposed to different GLA concentrations is shown. After 4 weeks of treatment, GLA caused a significant reduction in the growth rate at all tested concentrations.

In Figure 5, the trend of the number of eggs laid per week, for each GLA concentration, is shown, while in Table 1, the analytical data are reported. 

With respect to the negative control, GLA induced a significant reduction in the number of eggs laid at all tested concentrations. Interestingly, in Figure 5 we can observe a general reduction in the number of eggs laid followed by an increase in the last week. Such a trend is confirmed by the G-test correlation coefficient, which drastically decreases for the last week, indicating that groups tend to have more similar egg numbers among treatments.

Our PCA (Figure 6) clearly discriminates the negative control from the highest-concentration groups, confirming what was previously described by the other variables. 

## 4. Discussion

One of the consequences of the intensive use of herbicides is the discharge of residues into the environment, which contaminates the ecosystem and impacts the food chain with observable repercussions on the biota and on human health [21,22]. 

Among herbicides, GLA is one of the most used in agriculture practices, mainly to control weeds in genetically modified crops [8,9]. Therefore, its sublethal adverse effects and its ever-increasing use over the past decades represent a serious concern. 

Few data are present in the literature about the possible association of GLA with increased levels of genomic damage. Moreover, in most toxicology studies, the exposure concentrations are much higher than the levels found in environmental samples [11,18,20]. 

In the present paper, we filled these gaps by evaluating the ability of the GLA herbicide to induce, at environmentally relevant concentrations, the formation of MNi in vitro on cultured human lymphocytes and in vivo on hemocytes of *L. stagnalis*.

The results of our in vitro study provided evidence of the genotoxic effects of GLA on human lymphocytes at concentrations of 0.500 μg/mL down to 0.100 μg/mL (Figure 1 and Appendix A), indicating that, at these concentrations, GLA could pose genotoxic risk to humans, as also confirmed by other authors [29]. 

Although the mechanisms underlying the genotoxic potential of GLA are unknown, oxidative stress appears to play an important role in this regard. In fact, it is known that many pesticides are capable of inducing an increase in free radicals, whose damage at the cellular and genomic level has been known for some time [43,44].

The genomic instability was also evaluated by the frequency of NBUDs, which represent the elimination process of amplified DNA or excess chromosomes from aneuploidy cells. In this case, no significant differences were found at all tested concentrations, indicating that, at these levels, GLA does not affect the levels of NBUDs. 

A valid parameter used to determine the cytotoxic potential of xenobiotics is the mitotic index (MI), which represents a critical index in determining the rate of cell division. The decrease in MI indicates a slower cell progression from the S to M phase of the cell cycle, and, therefore, represents an index of cytotoxic damage resulting from exposure to xenobiotics.

In our study, contrary to what we showed for glyphosate [40], for the CBPI, we observed a significant effect of GLA at all tested concentrations, with the only exception of 0.010 μg/mL. This result indicated that exposure to GLA could lead to increased cytotoxic effects. However, at the ADI concentration, GLA did not influence the replicative capacity of the cells. 

Other authors observed decreased levels of the proliferation index in cell lines exposed to other herbicides or insecticides, but at concentrations higher than those tested in the present paper [45]. Moreover, a wide range of pesticides have been shown to induce cell damage, including those related to the cell cycle and cell growth. For example, the herbicide pendimethalin was found to induce cell cycle arrest and apoptosis in TM3 and TM4 cells by activating the endoplasmic reticulum stress pathway and autophagy [46]. Analogously, glyphosate has been reported to induce cell arrest in the G_0_/G_1_ phase [47], while exposure to sublethal doses (10, 20, and 40 mg L^−1^) of 2,4-dichlorophenoxyacetic acid resulted in a significant reduction in the MI [48].

A brief overview of the GLA biochemistry in animal cells can explain the origin of the observed genomic instability and reduced cell proliferation. As reported above, GLA’s structural similarity to glutamate (Figure 7) enables it to inhibit GS activity in plant cells, resulting in ammonia overaccumulation and ROS production [1]; nevertheless, in animal cells, the inhibition of GS, still observed, can be partially compensated through alternative metabolic pathways, preventing significant disruptions in ammonia metabolism and more severe physiological outcomes. In particular, aquatic organisms are generally ammonotelic and are therefore able to directly excrete NH_3_; nitrogen fixation in uric acid or urea is another possible strategy exploited to expel it from the organism [49]. Furthermore, the glyoxylate derivatives’ overaccumulation and consequent ROS production observable in plants derive from the relationship between GS with RuBisCO [3]. Since the Calvin cycle does not concern animals, the generation of ROS and subsequent necrosis is avoided. While the biological implications of GLA exposure are remarkably different between plant and animal cells, the basic biochemical interaction mechanisms between GLA and GS are similar. GS blockade leads to a decrease in the intracellular glutamine level, which is essential for cell growth and proliferation [50,51,52,53]. Glutamine depletion is in fact associated with reduced levels of β-TrCP (beta-transducin repeat-containing protein), a protein involved in timely mitosis regulation through targeted protein stability modulation (e.g., Emi1) [54]. The absence of β-TrCP leads to anomalies potentially connected to genomic instability and poor cell division, such as a lengthened mitosis, incorrect chromosome segregation, centrosome overduplication, and multipolar metaphase spindles [55,56].

The potentially negative effects of GLA on aquatic organisms are largely unknown. The great diversity of invertebrates, associated with their widespread distribution, usually exposes them to various levels of pollutants, making these organisms useful models in ecotoxicological studies. In our work, on the aquatic invertebrate *L. stagnalis,* we tested in vivo the same concentrations of GLA used for in vitro experiments. This organism has a short life cycle and is easy to rear in the laboratory [35], making it useful and convenient for chronic ecotoxicological studies. We found genotoxic properties of GLA at the highest concentrations of 0.500 and 0.200 μg/mL, whereas at the ADI concentration value of 0.010 μg/mL, GLA did not show significant differences in the level of MNi and NBUDs with respect to the negative control, confirming the 0.010 μg/mL value as a safe concentration for this compound. Data about the genotoxic properties of GLA are conflicting. Indeed, while some published papers evidenced the ability of GLA to induce cell death, chromatin condensation, abnormalities in the neuroepithelium of mouse embryos [57], and increased levels of MNi in gill cells of fish [30], others reported negative results [29]. 

Moreover, our results are comparable with literature data for *L. stagnalis* exposed to other xenobiotics such as Benzophenone-3, for which genotoxicity was observed at concentrations of 0.200 and 0.100 mg/L [38], and glyphosate, for which we observed a significant increase in MNi frequency at 0.500 and 0.250 µg/mL [58].

Finally, the greatest negative effects of GLA treatments were observed in terms of the number of eggs laid and body growth, which was significantly reduced at all tested concentrations. The significant reduction in body growth observed in our study agrees with the data obtained by Zhang et al. (2017) [59], who observed that GLA at concentrations of 5 and 10 μg/mL is able to inhibit the growth of *Microcystis aeruginosa.* The number of eggs laid is an important physiological indicator for the evaluation of reproductive capacity. Our results are consistent with data reported by Ahn et al. (2001) [60], who observed a significant larval and nymph death induced by GLA in *Orius strigicollis* and *Harmonia axyridis*, as well as with those reported by Zhao et al. (2023) [1], who observed, in *Caenorhabditis elegans*, significant reductions in the number of eggs and offspring in vivo and a significant increase in apoptosis in gonadal cells induced by treatment with different concentrations of GLA. 

## 5. Conclusions

This study is the first one investigating the ability of GLA to induce genomic instability in cultured human lymphocytes. We tested GLA concentrations ranging from 0.010 (ADI value) to 0.500 μg/mL, which represent more realistic concentrations, likely to be encountered in everyday life, than the higher doses evaluated in other published reports. 

We provided evidence of the genomic instability induced by GLA on cultured human lymphocytes and in vivo on *L. stagnalis* hemocytes. Despite the limitations of an in vitro study conducted on human lymphocytes and an in vivo study conducted on an invertebrate organism, it is our opinion that the increase in cytogenetic damage observed at GLA concentrations equal to or greater than 0.100 μg/mL needs attention, mainly in a chronic exposure scenario. Indeed, genomic instability caused by low but chronic exposure to genotoxic compounds can cause unexpected and serious consequences in the long run. 

Moreover, it should be emphasized that most pesticides, GLA included, are used in commercial formulations that contain inert compounds that are sometimes even more genotoxic than the active ingredients themselves. For example, the same commercial formulation of a GLA-based herbicide has been found to induce the formation of MNi in *Rhinella arenarum* tadpoles, in contrast to the active ingredient [9]. In a “One health” perspective, works like ours serve to highlight the dangers, both for human health and for the biome, of the chronic use of some pesticides at concentrations considered safe. This should lead the competent authorities to update the data relating to the acceptable daily intake dose for humans and those considered not harmful for the entire biome. In a future perspective, measuring cell growth could provide information on how cells respond to genotoxic treatments, and assessing the number of eggs hatched after pesticide treatment could evidence the possible effects of glufosinate across generations.

## Figures and Tables

**Figure 1 cells-13-00909-f001:**
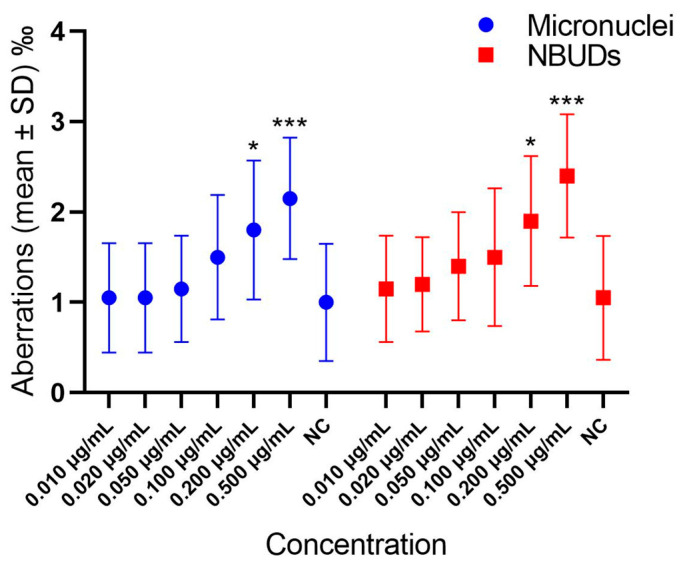
Graph highlighting the differences in the levels of micronuclei and nuclear buds in lymphocytes exposed to different concentrations of GLA. MMC = Mitomycin C (positive control); NC = negative control; NBUDs = nuclear buds. *** *p* < 0.001, * *p* = 0.011, with respect to negative control.

**Figure 2 cells-13-00909-f002:**
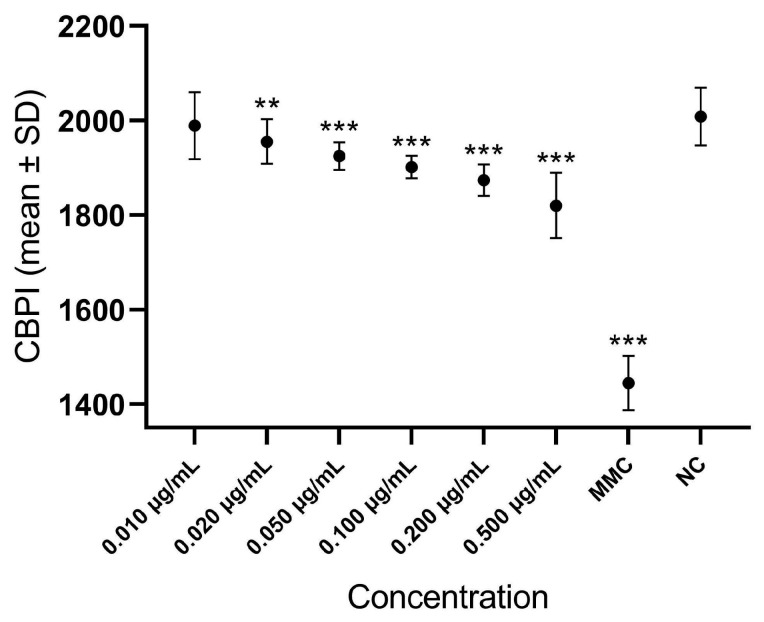
Differences in the CBPI values among lymphocytes treated with different GLA concentrations. CBPI = cytokinesis-block proliferation index; MMC = Mitomycin C (positive control); NC = negative control. *** *p* < 0.001, ** *p* = 0.002, with respect to negative control.

**Figure 3 cells-13-00909-f003:**
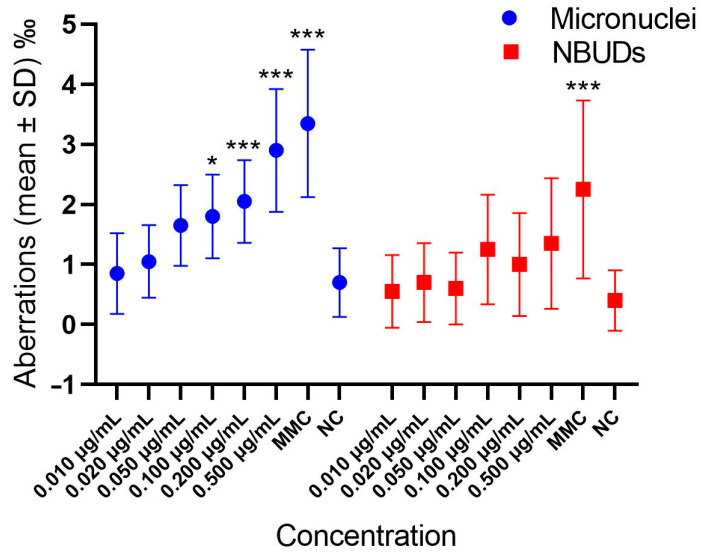
Frequencies of MNi and buds in hemocytes exposed to different GLA concentrations. NC = negative control; NBUDs = nuclear buds. Micronuclei: *** *p* < 0.001; * *p* = 0.023, with respect to negative control. NBUDs: *** *p* < 0.001, with respect to negative control.

**Figure 4 cells-13-00909-f004:**
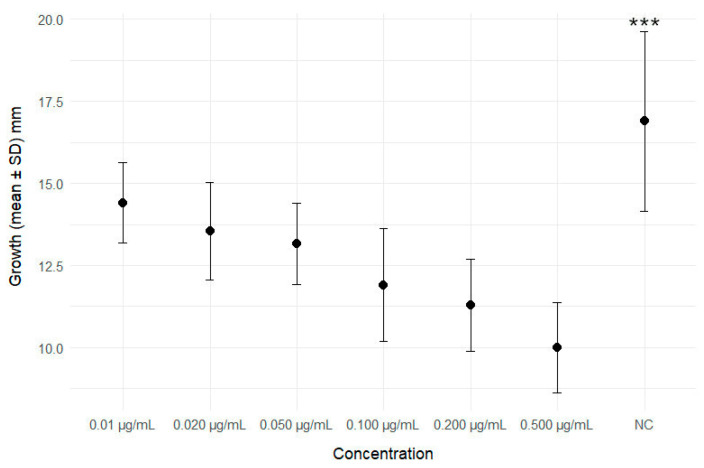
Variation in growth between controls and different GLA concentrations. NC = negative control. *** *p* < 0.001, significantly higher with respect to all GLA concentrations.

**Figure 5 cells-13-00909-f005:**
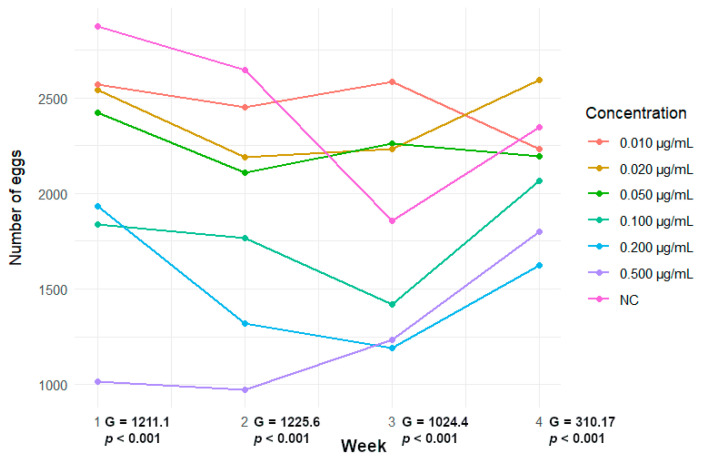
Trend of the number of eggs laid per week, for each GLA concentration.

**Figure 6 cells-13-00909-f006:**
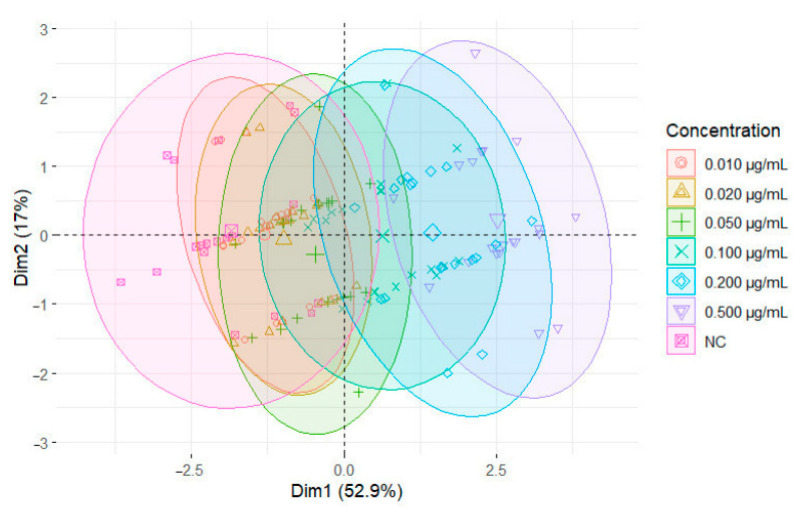
Principal component analysis (PCA) of our genotoxicity results, including the variables “Micronuclei”, “Nuclear buds”, “Binucleated cells”, “Growth”, and “Number of laid eggs”. The first two principal components account for 52.9% and 17.1% of the variance, respectively. Ellipses represent 95% confidence regions for the concentration groups.

**Figure 7 cells-13-00909-f007:**
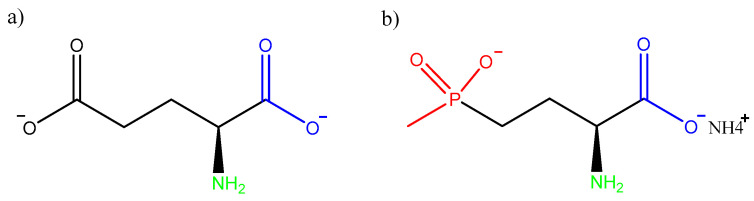
(**a**) L-glutamate and (**b**) GLA (L-phosphinothricin ammonium); the molecular structure similarity between the two molecules allows them to take part in the same metabolic pathways. The functional group (methylphosphonyl) that gives GLA its xenobiotic characteristic is highlighted in red: after its phosphorylation by ATP, it binds irreversibly to GS. Common functional groups of the two molecules that contribute to the GS binding through H-bonds are highlighted in blue and green [4].

**Table 1 cells-13-00909-t001:** Analytical data about the number of eggs laid per week per concentration after GLA treatment at different concentrations. Number of subjects tested per concentration = 20.

Concentration	I Week	II Week	III Week	IV Week	Total
NC	2875	2644	1856	2348	9723
0.500 µg/mL	1017	975	1235	1801	5028 ***
0.200 µg/mL	1934	1318	1190	1623	6065 ***
0.100 µg/mL	1838	1765	1421	2067	7091 ***
0.050 µg/mL	2423	2108	2258	2195	8984 ***
0.020 µg/mL	2542	2189	2232	2595	9558
0.010 µg/mL	2571	2449	2584	2232	9836

NC = negative control; *** *p* < 0.001; significantly lower with respect to NC (G-test).

## Data Availability

The data presented in this study are available in results and Appendix A sections.

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
