# Peer review of "Cellular and Genomic Instability Induced by the Herbicide Glufosinate-Ammonium: An In Vitro and In Vivo Approach"

_cells, 2024, doi:10.3390/cells13110909_

Round 1

Reviewer 1 Report

Comments and Suggestions for Authors

Thanks for giving me the opportunity to review this work. This paper presents a study that aims to evaluate the toxicity of glufosinate-ammonium on human cells and an invertebrate animal. I am not an environmental expert, but I feel the paper is fluent generally. The logic is very direct. The text is easy to read. The results can be expected. I only have the following four main suggestions for authors, and I hope they are helpful to improve the quality of the paper. First, the Introduction section should be reconstructed. There are too many paragraphs, and this will cause some confusion for understanding your logic. For example, when I first read the Lymnaea stagnalis in Line 92, I feel weird why you choose this organism. Then, you mentioned this point in the Line 112. Still, I am interested in why you do not use model fish (such as zebrafish) to conduct their experiments. I suggest that authors separately construct a paragraph to elucidate the reasons why they choose human cell and invertebrate organism as their target. Secondly, authors measured a series of indicators, but I am confused what correlations exist between these indicators. Please give explanations in proper place in the manuscript in detail. Thirdly, the results seem very simple, clear, and unsurprising. For me, this is a question. I think I just read a report but not a study that has been elaborately designed. Finally, please highlight the practical meanings of your results and present more discussions for the political suggestions.

Author Response

REVIEWER 1

Reviewer 1: Thanks for giving me the opportunity to review this work. This paper presents a study that aims to evaluate the toxicity of glufosinate-ammonium on human cells and an invertebrate animal. I am not an environmental expert, but I feel the paper is fluent generally. The logic is very direct. The text is easy to read. The results can be expected.

Authors: Thank you for appreciating our work

 Reviewer 1. I only have the following four main suggestions for authors, and I hope they are helpful to improve the quality of the paper.

Authors: Thank you for your suggestions that certainly improve our work

 Reviewer 1. First, the Introduction section should be reconstructed. There are too many paragraphs, and this will cause some confusion for understanding your logic. For example, when I first read the Lymnaea stagnalis in Line 92, I feel weird why you choose this organism. Then, you mentioned this point in the Line 112.

Authors: we have merged the two paragraphs regarding Lymnaea stagnalis

 Reviewer 1. Still, I am interested in why you do not use model fish (such as zebrafish) to conduct their experiments. I suggest that authors separately construct a paragraph to elucidate the reasons why they choose human cell and invertebrate organism as their target.

Authors: We understand that fish, and in particular zebrafish, are widely used in this type of experiment. However, each model organism has its own advantages and disadvantages. Lymnaea stagnalis, in addition to being still little studied, has a series of advantages linked to both reproduction (having a high egg production rate) and to the non-invasiveness of the sampling nature. As suggested, we included in the Introduction session, a paragraph that elucidated the reasons why we chose this two cell systems. In particular, we stated:

“Among the target cells used in literature to evaluate the effects of exposure to genotoxic agents, lymphocytes represent the ideal biological dosimeters for a variety of reasons: a) they are cell that circulates throughout the body and, therefore, can be considered circulating dosimeters able to detect the genotoxic insult received from the various districts of the organism; b) they can be obtained by a common venous sampling and are easy cultured in vitro; c) finally, the lymphocyte memorizes the damage, and, therefore, if the chromosomal damage persists unrepaired in cells in the G0 phase, the same can be detected when the cell undergoes mitotic division in vitro. Therefore, an increased frequency of MN may be the result of recent exposures, or, in some cases, of previous exposures, and therefore the manifestation of a cumulative effect (Ludovici et al., 2021).

 References

  1. M. Ludovici,M. G. Cascone, T. Huber, A. Chierici, P. Gaudio, S. O. de Souza, F. d’Errico, A. Malizia.Cytogenetic bio-dosimetry techniques in the detection of dicentric chromosomes induced by ionizing radiation: A review. Eur. Phys. J. Plus (2021) 136:482. https://doi.org/10.1140/epjp/s13360-021-01447-3

"L. stagnalis is often used as a model organism in ecotoxicological and biological studies, as it has different advantages: a) it is sexually mature within only 3 months after egg laying; b) it is a simultaneous hermaphrodite, which means that it can both cross- and self-mate; c) during mating, partners regularly alternate sexual roles by reciprocal exchange of sperm. These characteristics offer the possibility to study the influence of xenobiotic compounds on life-traits relating to the reproductive aspects, such as number of laid eggs; d) L. stagnalis can be easily maintained in the laboratory; it tolerates a wide range of temperatures (from 0 to 26–28 °C) and pH (from pH 6 to pH 8.5); e) this organism does not require a particular diet because it is omnivorous and feeds mainly on algae and plants; f) finally, L. stagnalis lives mainly in ponds, where the concentration of xenobiotics is higher than in other aquatic ecosystems. This means that they have probably evolved efficient cellular detoxification mechanisms and, therefore, the genotoxic effect of a certain environmental xenobiotic observed on this organism could have even worse consequences in other aquatic and terrestrial organisms living in contaminated ecosystems (Santovito et al., 2023)”.

 References

Santovito A, Pappalardo A, Nota A, Prearo M, Schleicherová D Lymnaea stagnalis and Ophryotrocha diadema as Model Organisms for Studying Genotoxicological and Physiological Effects of Benzophenone-3. Toxics. 2023, 30;11(10):827. doi: 10.3390/toxics11100827.

 Reviewer 1. Secondly, authors measured a series of indicators, but I am confused what correlations exist between these indicators. Please give explanations in proper place in the manuscript in detail.

Authors: MNI, NBUDS and CBPI are evaluated as part of the same test, the CBMN test. However, while MNis represent a consequence of chromosomal damage, NBUDs represent the manifestation of a process of elimination of a DNA amplified in excess and/or of an excess of DNA repair complexes and chromosomes. These concepts were explained in the introduction. Finally, the CBPI represents a cytokinesis block proliferation index and is calculated according to the following formula: [1 × N1] + [2 × N2] + [3 × (N3 + N4)]/N, where N1– N4 represents the number of cells with 1-4 nuclei, respectively, and N is the total number of cells evaluated. This index gives an idea of the influence of a specific xenobiotic on the cell's ability to divide: low values of this index correspond to a lower ability of the cell to divide, and this may represent the cellular manifestation of possible damage to the cytokinetic mechanism.

We have added this consideration to the bottom of the introduction.

Reviewer 1. Thirdly, the results seem very simple, clear, and unsurprising. For me, this is a question. I think I just read a report but not a study that has been elaborately designed.

Authors: While the results might appear straightforward, their implications are significant, especially considering the growing concerns about the environmental and health impacts of Glufosinate-ammonium. Our findings contribute valuable insights into the genotoxicity of Glufosinate-ammonium at environmentally relevant concentrations, which have not been extensively documented. This is critical as it directly relates to public health and safety regulations.

 Reviewer 1. Finally, please highlight the practical meanings of your results and present more discussions for the political suggestions.

Authors: We think that in a "One health" perspective, works like ours serve to highlight the dangers, both for human health and for the biome, of the chronic use of some pesticides at concentrations considered safe. This should lead the competent authorities to update the data relating to the acceptable daily intake dose for humans and those considered not harmful for the entire biome.

We included these considerations in the conclusion session.

Reviewer 2 Report

Comments and Suggestions for Authors

Comments on the Quality of English Language

The reviewer found no difficulty understanding the manuscript. The writing appears to be clear. 

Author Response

REVIEWER 2

Reviewer 2: The choice of the study subjects, including cultured human cells and organisms that could be exposed to GLA in nature, represents two types of major threats posed by the abusive use of herbicides. The data and its presentation are generally clear. However, the scope of the study, including the limited types and number of assays performed by the authors, do not seem to provide a clear and convincing picture that GLA, present at higher than recommended concentrations, poses a major problem. The authors should provide additional evidence to enrich their manuscript.

Authors: Our research not only utilised in vitro assays on cultured human lymphocytes but also included in vivo experiments on Lymnaea stagnalis hemocytes, making it more comprehensive than typical studies that focus on a single model system. The use of both human lymphocytes and an invertebrate model allows us to explore the effects of glufosinate-ammonium (GLA) across different biological systems, giving a broader assessment of its potential risks. Moreover, our methodology included multiple concentrations and endpoints, such as micronuclei formation and cytokinesis-block proliferation index, to thoroughly evaluate GLA's genotoxicity and cytotoxicity. Additionally, our study extends beyond the examination of genotoxicity and cytotoxicity by incorporating ecological and reproductive aspects, specifically the effects of GLA on egg production and growth in Lymnaea stagnalis. This inclusion offers insights into the broader ecological impacts of GLA, and particularly its effects on reproductive health and population dynamics in non-target aquatic organisms.Together, these factors of our research (examining multiple biological models, incorporating various endpoints, and assessing ecological impacts) strengthen the manuscript by providing a multi-dimensional analysis of GLA's risks. This comprehensive approach highlights the potential broader implications of GLA exposure in the environment and supports the case for further regulatory review and risk assessment.

Major comments:

Reviewer 2: The reviewer does not understand the term “genomic damage” used by the authors throughout the manuscript. Do the authors mean damage in the form of DNA damage? However, none of the data presented in the manuscript can be considered as direct or indirect measurements of DNA damage. Instead, phenotypes such as micronuclei formation or other nuclear abnormalities can be considered as signs of genome instability. The authors should consider changing the use of terms and be more precise.

Authors: We agree. In the text, we substituted “genomic damage” with “genomic Instability”.

 Reviewer 2: Rearing conditions of L. stagnalis. The authors should provide evidence that the tab water and the salad used to feed L. stagnalis are free of GLA given how widespread the compound is. Otherwise the design of the study could be flawed.

Authors: In the revised version of the paper, we added information about the absence of GLA in the tap water and salad used during the experiments. In particular, in the material and method session, Lymnaea stagnalis physiology and rearing subsession, we added the following sentence: “The chemistry of the used tap water was: pH: 7.4; dry residue at 180 °C: 307 mg/L; calcium: 69 mg/L; magnesium: 13 mg/L; ammonium: <0.05 mg/L; chlorides: 14 mg/L; sulfates: 32 mg/L; potassium: 2 mg/L; sodium: <10 mg/L; arsenic: <1 µg/L. Glufosinate and other pesticides were not found (Smat Torino, 2023)”. Moreover, “In order to avoid using salad treated with pesticides, we fed L. stagnalis with organic salad”.

We would like to emphasize that, although it is not possible to have a complete guarantee that what is indicated as organic does not have traces of pesticides, it must be considered that both the controls and the treated subjects were raised with the same water, the same food, and in the same room. Therefore, also in case of possible traces of pesticides in the water or salad, this fact would not have compromised the validity of the results.

Reference:

SMAT Turin. Quality characteristics of the water supplied in the municipality of Turin, TO 1 District (Average Values, Second Half of 2023). Available online: https://www.smatorino.it/monitoraggio-acque/ (accessed on 10 May 2024).

Reviewer 2: Figure 1 & 3, and Supplementary Tables 1 & 2. In Figure 1 & 3, it was not immediately clear what unit was used to describe the levels of aberrations. Upon examining the data in Supplementary Tables, % is likely the unit. However, using the formula listed in the tables, the calculations appeared wrong – the authors wrongfully placed decimal points. i.e. in Table 1 in the MMC-treated samples, 67 MNi were observed out of 20,000 lymphocytes examined, resulting in a frequency of MNi/N = 67/20,000 = 0.335%, instead of 3.350%. This must be corrected.

Authors: it was a mistake. In the revised version, we substituted % with in Supplementary Tables 1 & 2. In Figure 1 and 3, we substituted “Aberration (mean ± DS)” with Aberration mean ± DS) .

 Reviewer 2. In Figure 2, the authors used cytokinesis-block proliferation index (CBPI), a measurement invented by the same first author and used in another publication to evaluate the growth of lymphocytes. Can the authors provide data demonstrating lymphocyte growth more conventionally, i.e. the growth curve of the cells?

Authors: In this case, we disagree with the Reviewer. The CBPI index is an index commonly calculated in the micronucleus assay (Some examples in Masubuchi et al., 2003; Yilmaz et al., 2023), and does not represent a measure of cell growth, but rather represents a cytokinesis block proliferation index. It is calculated with the following formula: [1 × N1] + [2 × N2] + [3 × (N3 + N4)]/N, where N1– N4 represents the number of cells with 1-4 nuclei, respectively, and N is the total number of cells evaluated. This index gives an idea of the influence of a specific xenobiotic on the cell's ability to divide: low values of this index correspond to a lower ability of the cell to divide, and this may represent the cellular manifestation of possible damage to the cytokinetic mechanism.

We added this consideration to the bottom of the introduction.

Reference:

Masubuchi, S. Yamada, T. Horie, Possible mechanism of hepatocyte injury induced by diphenylamine and its structurally related nonsteroidal anti-inflammatory drugs, J. Pharmacol. Exp. Ther. 292 (2000) 982-987.

Ece Avuloglu Yilmaz, Deniz Yuzbasioglu and Fatma Unal. Investigation of genotoxic effect of octyl gallate used as an antioxidant food additive in in vitro test systems. Mutagenesis, 2023, 38(3): 151–159.

Reviewer 2: The authors should justify the use of G-test in Table 1, which suggests significant differences in L. stagnalis’s egg-laying capacities over time, especially given the total number of eggs, and the number of eggs produced per week was not much different between negative control and the treatment groups where GLA was used at 0.010, 0.020 and 0.050 μg/mL.

Authors: We made a mistake in Table 1. We revised Figure 5 indicating the p and G values for the G test applied on the eggs laid in the 4 weeks separately, and we revised Table 1 indicating which of the amounts of eggs laid at the 6 concentrations was significantly different from the control group.

 Reviewer 2. As shown in Figure 5, the observation that L. stagnalis appeared to tolerate GLA and recover its egg-laying capacities over time is interesting. To better demonstrate that the eggs, such as their quality were not negatively affected during prolonged GLA exposure, the authors should consider providing other evidence. For example, does prolonged GLA exposure lead to a lower percentage of hatches from the eggs?

Authors: our work focused mainly on the genotoxicological aspect, although the proposed approach would certainly complement that linked to egg production. However, it would take a long time for the eggs to hatch and the hatched individuals to grow to a size large enough to be seen and counted. Furthermore, in the current situation, including this parameter would mean completely repeating the experiment.

Reviewer 2. Line 359. A self-citation of unpublished data is inappropriate.

Authors: we agree. However, data have now just been accepted for publication in Aquatic Toxicology journal. Thus, we include the appropriate citation in the references session and in the main text.

Reference:

Dáša Schleicherová, Paolo Pastorino, Alessia Pappalardo, Alessandro Nota, Claudio Gendusa, Enrico Mirone, Marino Prearo, Alfredo Santovito. Genotoxicological and physiological effects of glyphosate and its metabolite, aminomethylphosphonic acid, on the freshwater invertebrate Lymnaea stagnalis. Aquatic Toxicology, 2024, 271: 106940

Minor comments:

Reviewer 2. Typo on line 202. Two “*” were used for P-values <0.01 & 0.001. However, throughout the manuscript, the authors did not seem to adhere to this rule they established. For example, in Table 1, the authors used a single “*” to demonstrate a P-value less than 0.001.

Authors: it was a mistake. In the revised version of the paper, the asterisks were indicated in a corrected manner.

Reviewer 2. Missing unit in Figure 4 and supplementary table 3. Though described in the material and methods section (on line 173), a change of shell length, as measured in mm, was used to evaluate the growth of L. stagnalis individuals, it is still recommended to show the unit on the y-axis of Figure 4 and in Table 3.

Authors: we modified Figure 4 and Supplementary table 3 adding the missing unit.

Reviewer 2. Line 295. The reviewer does not understand the statement that “GLA does not affect transcription mechanisms.” By saying transcription, do the authors mean RNA transcription? RNA transcription is one of the many factors that could contribute to nuclear abnormalities and genome instability. This statement by the authors is unclear and needs clarification

Authors: yes, considering that NBUDs represent the manifestation of the elimination process of amplified DNA, their appearance could be due to defects in transcription mechanisms. However, NBUDs also represent an elimination of excess chromosomes from aneuploidy cells. For this reason, in order to avoid confusion, in the revised version of the text we substituted “transcription mechanisms” with “the levels of NBUDs”.

Reviewer 3 Report

Comments and Suggestions for Authors

The authors investigated the genotoxicity of the herbicide Glufosinate-ammonium (GLA)  on cultured human lymphocytes and Lymnaea stagnalis hemocytes at six concentrations: 0.010 (the established acceptable daily intake value), 0.020, 0.050, 0.100, 0.200 and 0.500 µg/mL using in vitro and in vivo micronucleus assays. their results show a significant and concentration-dependent increase of micronuclei frequency at concentrations from 0.100 to 0.500 µg/mL in human lymphocytes, while in L. stagnalis hemocytes, significant differences were found at 0.200 and 0.500 µg/mL. A significant reduction of the proliferation index was observed at all tested concentrations, with the only exception of 0.010 µg/mL, indicating that the exposure to GLA could lead to increased cytotoxic effects. In L. stagnalis, significant reduction of laid eggs and body growth were also observed at all concentrations. They conclude that GLA has the potential to cause genomic and cellular damage

 The paper is well written and easy to follow. The data is well presented I only have a few minor edits to recommend.

The paper is well written and easy to follow. The data is well presented I only have a few minor edits to recommend.

Line 67 – Potential environmental concentration in Canada as 1 mg/mL seems high. I know the value was correctly stated from a reference. I was not able to find a different value on the PMRA website for that pesticide.

 Line 101 – You have “resulting visible as extranuclear bodies”. Should be “resulting in visible extranuclear bodies” or “resulting  as visible extranuclear bodies”

 Line 177 – forgot the units after the list of concentrations

 Line 286 – not sure if the units in µg/mL is what you wanted as you refer to the potential concentration in Canada in mg/mL in line 67. However the value stated in line 286 is one order of magnitude greater than 0.100 µg/mL.

Comments on the Quality of English Language

No real concern with the English language it is very good and easy to follow. I only see a couple of minor grammar edits.

Author Response

REVIEWER 4

The authors investigated the genotoxicity of the herbicide Glufosinate-ammonium (GLA) on cultured human lymphocytes and Lymnaea stagnalis hemocytes at six concentrations: 0.010 (the established acceptable daily intake value), 0.020, 0.050, 0.100, 0.200 and 0.500 µg/mL using in vitro and in vivo micronucleus assays. their results show a significant and concentration-dependent increase of micronuclei frequency at concentrations from 0.100 to 0.500 µg/mL in human lymphocytes, while in L. stagnalis hemocytes, significant differences were found at 0.200 and 0.500 µg/mL. A significant reduction of the proliferation index was observed at all tested concentrations, with the only exception of 0.010 µg/mL, indicating that the exposure to GLA could lead to increased cytotoxic effects. In L. stagnalis, significant reduction of laid eggs and body growth were also observed at all concentrations. They conclude that GLA has the potential to cause genomic and cellular damage

The paper is well written and easy to follow. The data is well presented I only have a few minor edits to recommend.

Authors: thank you for appreciating our work

Reviewer 4. Line 67 – Potential environmental concentration in Canada as 1 mg/mL seems high. I know the value was correctly stated from a reference. I was not able to find a different value on the PMRA website for that pesticide.

Line 286 – not sure if the units in µg/mL is what you wanted as you refer to the potential concentration in Canada in mg/mL in line 67. However the value stated in line 286 is one order of magnitude greater than 0.100 µg/mL.

Authors: Effectively, we took this concentration value from a published paper (Zhao, X. et al., 2023). However, we were not able to find confirmation of this threshold on the Canadian PMRA website or in other published papers. Moreover, the above-mentioned publication refers to a paper dated 1998 who reported data of “eutrophic” Canadian lakes. For this reason, in the revised version of the paper, also considered the little relevance of this datum in the main context of our paper, we decided to delete it.

 Reviewer 4.  Line 177 – forgot the units after the list of concentrations

Authors: in the revised version, we added the units after the list of concentrations.

Round 2

Reviewer 1 Report

Comments and Suggestions for Authors

Thanks for the authors' response. I have no further comments.

Author Response

RESPONSE TO REVIEWER 1

Reviewer 1: Thanks for the authors' response. I have no further comments.

Authors: thank you for your favorable opinion about the work.

Reviewer 2 Report

Comments and Suggestions for Authors

The authors were able to sufficiently address most of the comments and corrected errors.  See attachment.

Comments on the Quality of English Language

The reviewer did not have difficulty in understanding the presentation. 

Author Response

RESPONSE TO REVIEWER 2

Reviewer 2 - Additional reviewer response:
The reviewer cannot find any information about CBPI in the new reference article provided by the authors (Masubuchi et al., 2000). Besides, the “year” of the article mentioned by the authors in the response is 2003, which does not match the listed reference. Can the authors make sure they are citing the right paper?
In general, please cite the original paper where the CBPI index is introduced instead of self-citation.
Authors: we apologize, in the revised version of the paper we included the following original citation instead of the self-citation:

Surrallés, J.; Xamena, N.; Creus, A.; Catalfin, J.; Norppa, H.; Marcos, R. Induction of micronuclei by five pyrethroid insecticides in whole-blood and isolated human lymphocyte cultures. Mutat. Res. 1995, 341, 169-184.

Reviewer 2 - In addition, the reviewer was asking for a different point that the authors failed to respond to. The reviewer understands that CBPI is not a not a direct measurement of cell proliferation. But instead of only measuring phenotypes associated with genome instability such as CBPI, it is often worthwhile to provide additional information, such as a measurement of how cells respond to the genotoxic treatment in a proliferation assay.

Authors: We understand the reviewer’s request for a conventional growth curve to directly demonstrate lymphocyte proliferation. Unfortunately, we do not have additional data on cell growth curves for lymphocyte proliferation under the same experimental conditions, and it is now impossible for us to provide them. This would have been an interesting addition to our study. Our work, however, includes several other fitness outputs, such as shell growth, micronuclei frequency, and number of eggs laid, providing an already comprehensive assessment of the effects of GLA.

Reviewer 2: Additional reviewer response:
The reviewer understands that looking into the hatched offsprings will take more time and therefore out of the scope of the current study. But it will be interesting to see if GLA’s effect can be transmitted across generations. If so, this will potentially significantly increase the impact of the current study

Authors: We agree with the reviewer that investigating whether GLA’s effects can be transmitted across generations would be a valuable and impactful improvement of the current study. However, as the reviewer correctly notes, examining hatched offspring and assessing transgenerational effects would require a significantly longer timeframe and is beyond the scope of the current study. Future research could explore this aspect to provide insights into the long-term and multigenerational impacts of GLA exposure.For this reason we decided to include this consideration in the conclusion session.